# Limited Angle Electrical Resistance Tomography in Wastewater Monitoring

**DOI:** 10.3390/s20071899

**Published:** 2020-03-29

**Authors:** Chenning Wu, Martin Hutton, Manuchehr Soleimani

**Affiliations:** 1Engineering Tomography Laboratory (ETL), Department of Electronic and Electrical Engineering, University of Bath, Bath BA2 7AY, UK; cnw24@bath.ac.uk; 2Ashridge Engineering LTD., Okehampton EX20 1BQ, UK; m.hutton@ash-eng.com

**Keywords:** limited data ERT, limited region ERT, part-filled pipe monitoring

## Abstract

Electrical resistance tomography (ERT) has been investigated in monitoring conductive flows due to its high speed, non-intrusive and no radiation hazard advantages. Recently, we have developed an ERT system for the novel application of smart wastewater metering. The dedicated low cost and high-speed design of the reported ERT device allows for imaging pipes with different flow constituents and monitoring the sewer networks. This work extends the capability of such a system to work with partially filled lateral pipes where the incomplete data issue arises due to the electrodes losing contact with the conductive medium. Although the ERT for such a limited region has been developed for many years, there is no study on imaging content within these limited regions. For wastewater monitoring, this means imaging the wastewater and solid inclusions at the same time. This paper has presented a modified ERT system that has the capacity to image inclusions within the conductive region using limited data. We have adjusted the ERT hardware to register the information of the non-contact electrodes and hence the valid measurements. A limited region image reconstruction method based on Jacobian reformulation is applied to gain robustness when it comes to inclusion recovery in limited data ERT. Both simulation and experimental results have demonstrated an enhanced performance brought by the limited region method in comparison to the global reconstruction.

## 1. Introduction

Urban wastewater is defined in the Directive as the mixture of domestic wastewater, the wastewater from industries discharging to sewers and rainwater run-off from roads and other impermeable surfaces. Effective wastewater treatment is designed to remove various contaminants of sewage solids, pathogens, nutrients, toxic chemicals and metals so that treated wastewater can be returned to the environment. The other by-product of sewage treatment is sewage sludge, which needs to be appropriately re-utilized or disposed of [1]. A sewerage system is a network of pipes, pumping stations and appurtenances that convey sewage from its points of origin to a point of treatment and disposal. It must accommodate for a wide variation of flow rates over the course of a day, especially for the peak flow rate, as flow quantities depend upon population density, water consumption and the extent of commercial or industrial activity in the community [2]. Due to the complexity of different sewerage systems, especially the combined systems, flow monitoring is vital for early stage blockage detection and hence reducing the chances of overflow. 

Electrical resistance tomography (ERT) has been developed extensively for visualizing and understanding the concentration distribution and flow behavior within a process instrument [3,4,5,6]. It involves the measurement of the independent mutual impedance between electrode pairs and the reconstruction of cross-sectional images using measured data and suitable algorithm. A conventional ERT system consists of a set of electrodes evenly mounted around the periphery of the object. Mutual impedance is obtained by injecting currents from one pair of electrodes and taking voltage measurements from another pair of electrodes. Typically, ERT sensors are dominantly applied to aqueous-based fluids that possess continuous admittance; hence they are suitable for wastewater flow applications as the operating conditions for fluid conductivity are regulated as 50 to 1200 µS cm^−1^ [7]. Moreover, ERT systems offer high temporal speed and the ability of visualization of multi-phase flow. Therefore, they outperform other traditional non-tomographic flowmeters, e.g., ultrasonic Doppler velocity profilers, electromagnetic meters, Coriolis mass meters, and Venturi meters, and become good candidates for monitoring and analyzing sewage flow behaviors. 

In [8], it was advised that foul sewers and lateral drains should be designed to run at no more than 75% of pipe full conditions. In other words, in the free-flowing sewers, the pipes are mostly part full. In such cases, the electrodes above the liquid surface will lose their electrical contact with the sensing field, which causes a great phase-shift in response potentials on these electrodes. Since the data acquisition system (DAQ) of the ERT system neglects the phase shifts among all measurements, voltage data on these electrodes cannot reflect the true conductivity distribution. To address this problem, [9] developed a novel sensor design which uses a single conductive ring to replace the discrete electrodes to guarantee a continuous current excitation. However, the performance of the design relies on the fluid conductivity and thickness ratios; hence, in the case of sewage flow monitoring, where a large conductivity variation could exist, the reconstruction quality would decay. Reconstruction methods have been focused on extracting the conductive phase surface. For instance, [10] proposed a liquid level detection method that directly analyses raw voltage measurements collected from a conventional ERT hardware system to estimate the liquid levels; [11] estimated the free-surface in two-phase flow by using the boundary element method to formulate the forward problem and the iterative Levenberg–Marquardt method for inverse problem. However, these methods can only provide the information on water surface levels as suggested, but fail to visualize within the flow. [12] proposed two methods based on the valid dataset and the new sensitivity field to reconstruct the stratified flow in a traditional ERT system. Firstly, the electrodes that are above the liquid level are identified and any measurements that involve these electrodes are eliminated from the dataset. The remaining valid dataset is used for image reconstruction. However, this incomplete set of data will worsen the ill-posed nature of ERT as fewer measurements are available for the reconstruction. Previous studies have investigated the ERT reconstruction methods for solving the partial data and limited angle problems [13,14,15,16]. However, the problem assessed previously considered the energy distribution over the entire continuous domain caused by an electric current injection through a partially accessible boundary. However, the part full pipe problems essentially divide the domains into two subsets, as the electric currents cannot propagate beyond the conductive phase. [12] took advantage of such prior knowledge of the region where the continuous phase exists and proposed the limited region reconstruction method by reformulating the sensitivity matrix. The simulation results were presented to compare the reconstruction within the localized region with only using the valid dataset. The advancement of such reconstruction method has also been proven experimentally in [17], however, with a better prior knowledge of location in the medical application. 

In this work, we further developed the ERT device designed in [18] so that it could accommodate for incomplete data application at a low cost; and a comprehensive study of the advantages brought by Jacobian matrix reformulation will be quantitatively investigated using both numerical simulations and phantom experiments. Inclusions of different sizes will be added into the continuous phase at various locations. Position error (PE), amplitude response (AR), shape deformation (SD) [19], as well as correlation coefficient (CC), relative error (RE) and computation time will be compared to justify the advantages brought by applying the localised method. 

## 2. Modified ERT Hardware 

The high temporal and low spatial resolution nature of ERT offers an opportunity to capture the flow profile in sewer systems. In [18], a 16-channel ERT device was designed for wastewater flow monitoring applications, as shown in Figure 1a. The reported 14 frames/second data acquisition speed and the smallest detectable size being 0.04% of the phantom area allow for a successful motion tracking practice. Additionally, the compact design, which was prototyped at 14 × 7 × 6 cm, and the cost effectiveness make it preferable for mass deployment in the sewer network, and hence facilitate maintenance and minimize disruptions to the networks. 

Figure 1b presents an overview of the 16-channel ERT system architecture. It operates at 50 kHz and uses the adjacent driving and measuring mechanism, which produces an overall 208 measurements per frame, with a user configurable current injection over the range of 6 to 18 mA. Since the ERT technique only concerns the in-phase response, a peak detector was introduced in the system for capturing the in-phase component. Consequently, the phase shifts caused by the discontinuous phase is then ignored. 

### 2.1. Current Sensing Module

Since the traditional current-injecting-voltage-sensing mechanism has difficulty producing accurate measurements when the electrodes involved are present in the air, these data points will have to be eliminated from the dataset. To obtain the knowledge of the electrodes that are not submerged in the liquid, a current sensing module, as illustrated in Figure 2, is therefore added to the previous ERT hardware. Specifically, a current sensing resistor is included in the return loop of the current-stimulation circuit such that the current flowing through it in the form of a voltage drop is fed into the controller; hence the open circuit formed by any electrode of the injecting/receiving electrode pair that is exposed in the air will be acknowledged by proper thresholding. Here, 500 instead of 0 was chosen in the reception module, which includes a 12-bit Analogue to Digital Converter to account for any voltage drop across the resistance over transmission. Once the non-contact electrodes are determined, the measurements taken from any of these electrodes can then be considered as erroneous and automatically eliminated from one frame of data. 

### 2.2. Incomplete Dataset Model

In our 16-channel ERT device, with all electrodes being capable of providing informative measurements, there are $N_{M}=N\;\times \;(N\;-\;3)$ independent measurements, where N is the channel number of the ERT device. In the stratified flow applications, some electrodes will lose contact with the continuous phase, as illustrated in Figure 3, and thus fail to provide valid measurements to the reconstruction, in which case the dataset is incomplete. The number of independent measurements available in such system now drop to $N_{M}{=(N}_{v}{\;-\;2)(N}_{v}\;-\;3)/2$, where $N_{v}$ is the number of valid electrodes and $N_{v}\;<\;N$. 

The ERT image reconstruction problem is known as being ill-posed, as the problem is formulated using an ill-conditioned sensitivity (Jacobian) matrix. The singular value decomposition analysis of the Jacobian matrix can be introduced to evaluate the degree-of-ill-posedness of such problems [20]. According to the Picard condition, the number of singular values above the noise level of the measurements represents the amount of information that can be extracted. Evidently, the decrease in the available independent measurements makes the singular values decay much faster and therefore fewer singular values will carry relevant information in the noisy case. 

## 3. ERT Reconstruction 

Image reconstruction is performed by field electrical modelling for forward model and inversion algorithm to recover the conductivity distributions from boundary measurements. The mathematical model describing the electrical properties of the conductive field is solved for forward model [21]. For accurate ERT reconstruction, it is a prerequisite to build a model to simulate voltages at the boundary for a given conductivity distribution, which is known as the forward problem. The forward problem can be solved numerically by discretizing the domain into small elements to turn a continuous problem into a discrete problem. This is commonly solved by the finite element method (FEM). 

For stable and fast image reconstruction, a linearized inverse problem is solved using the Jacobian Matrix J and L2- norm regularizing penalty term, e.g., the Tikhonov regularization method. The system is based on time difference imaging, which means it reconstructs the changes in electrical conductivities Δσ from the differential voltages Δu obtained at time $t_{1}$ and $t_{2}$. Here, we consider the two sets of measurements $u_{i}$ and $u_{b}$ as the measurements taken before and after the insertion of inclusions. Accordingly, the inverse problem can be solved following Equation (1):(1)$$\Delta {\sigma =(J}^{T}{J+\gamma }^{2}{R)}^{-1}J^{T}\Delta u$$
where R is the regularization matrix and is based on the discrete Laplacian; $\gamma^{2}$ is the regularization parameter, which is empirically selected. This paper presents cases for static imaging mode, in dynamical imaging mode temporal based algorithms could be adapted [22].

Limited region method

The conventional algorithm reconstructs images over the entire domain without emphasis on the information in the region of interest (ROI). The inherent problem that an ERT system suffers is linked with the ill-conditioned sensitivity matrix. The incomplete datasets generated from partially-filled phantoms further increase the condition number and can result in the even more significant numerical errors in the reconstructed images. A strategy of using the valid measurements to reconstruct images restricted to a pre-defined ROI is introduced. Essentially, only the conductivity changes within the continuous phase is concerned, since the electromagnetic waves are not able to propagate into the non-conductive phase. Therefore, the ROI should be chosen equivalent to the conductive area as closely as possible. Given the gap existing between two adjacent electrodes and the uncertainty of the exact liquid level, the ROI boundary is defined as the lowest level of the invalid electrodes to ensure the coverage of all the potential conductivity changes under the water as demonstrated in Figure 4. 

The principal of limited region reconstruction is restricting the process to the ROI defined by the water level estimator. Therefore, the original domain Ω can be divided into two areas: one is the continuous phase area, denoted as $\Omega_{ROI}\subseteq \Omega$, where conductivity changes will be picked up in the voltage measurements (also known as the ROI); the other is the discontinuous phase area that theoretically $\sigma_{air}=0$. Then, the conductivity changes Δσ can be mapped to
(2)$$\Delta \sigma =\{\begin{array}{c} {\Delta \sigma }_{\mathrm{ROI}},\;\;x\in \Omega_{\mathrm{ROI}}\\ 0,\;\;\;x\in \Omega \setminus \Omega_{\mathrm{ROI}} \end{array}$$
Now the limited region reconstruction equation can be derived as:(3)$$\Delta {V=J}_{x\in \Omega_{\mathrm{ROI}}}{\Delta \sigma }_{\mathrm{ROI}}$$
where ΔV is the difference boundary measurements, $J_{x\in \Omega_{ROI}}$ is the reformulated Jacobian matrix. Here we have identified the conductivity changes with their finite elements approximations. 

By limiting the imaging area, which consequently reduces the number of unknown pixels, we could enhance the system robustness when the same amount of accessible measurements are used. Other than a better accuracy, a faster computation can also be achieved using the limited region method when compared with the global reconstruction due to the reduction in the number of pixels involved in the imaging process. 

## 4. Results and Analysis

### Simulation Study 

In this section, a series of numerical simulations were performed on a unit domain $\Omega =\{\left(x,y\right){:\;x}^{2}{+y}^{2}\;<\;1\}$ with 16 electrodes equally spaced around its circumference. We set the region $\Omega_{1}=\{\left(x,y\right):\;y\;<\;\mathrm{Level}\}$ represent the continuous (liquid) phase, where Level serves as the interface between the liquid and gas phases; and the conductivity value of region $\Omega_{1}$ is 1. As for the conductivity value of region $\Omega_{1}\backslash \Omega$, which is to simulate the gas phase, should theoretically be 0 ${Sm}^{-1}$; but this will lead to stimulation currents unable to be injected into the system and hence the simulation failure. Consequently, we set it to $1\times 10^{-4}$
${Sm}^{-1}$, which is small enough to be distinguished from the liquid phase.

In the single inclusion simulation tests, we created three different scenarios:
Case1: Level=0.5 and we apply an anomaly $\Omega_{2}=0.1$ to region $D1=\{\left(x,y\right){:\;x}^{2}+{\left(y+0.2\right)}^{2}\;<\;0.04\}$; Case2: Level=0.2 and we apply an anomaly $\Omega_{2}=0.1$ to region $D2=\{\left(x,y\right){:\;x}^{2}+{\left(y+0.4\right)}^{2}\;<\;0.04\}$.Case3: Level=−0.2 and we apply an anomaly $\Omega_{2}=0.1$ to region $D3=\{\left(x,y\right){:\;x}^{2}+{\left(y+0.6\right)}^{2}\;<\;0.04\}$.

The forward problem is numerically solved using a MATLAB toolkit, i.e., Electrical Impedance and Diffuse Optical Reconstruction Software (EIDORS) [21], to generate simulated boundary measurements, $v_{i}$ and $v_{h}$, and we added randomly generated 12 dB Gaussian noise to the simulated data. The reconstructions using global and limited region methods were then compared in Table 1. The apparent improvements of image qualities are seen especially with the existence of noise in the localized images, which justify the benefits of using such method. It is worth pointing out that in Case 3, where only seven electrodes are simulated submerged in the water, we would expect the worst image quality among these three scenarios. However, due to the space limitation, the anomaly can only be placed very close to the boundary; hence, recovering images using the limited region method in particular did not experience more difficulty than the other two cases. This is the major difference between the problem arising from part full pipe flow and the general incomplete data problems. Accordingly, water levels were only advised to decrease to half full in the following phantom experiments for a better justification of applying the proposed methods at various locations. 

In the multiple inclusion simulation model, two anomalies of $\Omega_{d}=0.1$ were defined by D within the unit domain under the two liquid levels:

$$D=\left\{\left(x_{1}{,y}_{1}\right){:(x}_{1}+0.4{)}^{2}+\;{\left(y_{1}+0.6\right)}^{2}\;<\;0.01;\;\left(x_{2}{,y}_{2}\right){:(x}_{2}\;-\;0.4{)}^{2}+{\left(y_{2}+0.6\right)}^{2}\;<\;0.01\right\}\;\mathrm{Level}\;1=0;\;\mathrm{Level}\;2=-0.38$$

Again, 12 dB Gaussian noise was added to the simulated boundary measurements as shown in Table 2. Naturally, the reconstructions involving multiple inclusions are more challenging than the single inclusion problems due to the increased complexity. The lack of measurements makes it even more difficult to solve the inverse problems particularly with the disturbance of added noises. The localised images give improved results with a better reveal of objects, whereas those of the global images are rather distorted, especially in the Level 2 simulation. 

## 5. Experimental Study

### 5.1. Experiment Set up

In this section, feasibility studies were carried out experimentally considering only 2-dimensional models. Phantom experiments were established in a horizontally placed 11 cm diameter cylindrical tank with 16 electrodes of 1.2 × 1 cm size equally spaced stainless-steel electrodes shown in Figure 5. The modified ERT device was used to collect measurements under the stimulation currents of 10 mA. 

#### 5.1.1. Reconstructions Based on Valid Datasets

As discussed previously, the existence of electrodes not in contact with conductive medium would introduce errors into the conventional ERT system. We established a current sensing module to locate the erroneous electrodes and eliminate the incorrect measurements accordingly. In this section, the necessity of identifying the valid datasets for reconstructions in the cases of part full pipe applications is justified by comparing images of inclusions recovered from the valid against those from the full datasets. 

Experiments were conducted by collecting the background datasets with the tank full and second datasets after inserting a small and medium inclusion under five different liquid levels, as shown in Table 3 and Table 4. The real distributions of the phantoms are also provided for reference. 

Evidently, with no knowledge of erroneous measurements, the inaccurate measurements can seriously distort the images, especially in the small inclusion tests. On the other hand, the valid datasets managed to recover both the gas void and the inclusion in all tests. 

#### 5.1.2. Investigation on Limited Region Method

With the confidence of reconstructions using incomplete datasets, a further investigation on the advantages of applying prior knowledge of the conductive phase area will be discussed in this section. 

Three different water levels were considered in both single and multiple inclusion tests: Level 1: Electrodes 4–14 are submerged in the water;Level 2: Electrodes 5–13 are submerged in the water;Level 3: Electrodes 5–12 are submerged in the water,
where the electrode numbering is referred to in Figure 4.

As suggested previously, sewers are advised to run at part full conditions under normal operation; thus the full tank data are mostly inaccessible. To model such circumstances, we conducted the experiments using datasets taken before and after the insertion of the anomalies in the part full pipes. The influence of liquid level increase caused by the insertions was also ignored to simplify the problem and focus on recovering targets within the conductive phase. That is to say, the liquid level is kept the same for both before and after adding the inclusions to make sure conductivity changes are entirely generated by the addition of targets. Reconstructions were completed and compared between the global method and the limited region method. 

Single inclusion test

In all cases, small (2 cm diameter) and medium (3 cm diameter) plastic rods were placed at various locations under the water within the pipe. 

The reconstructed images of two objects using the global and localised methods (ROI) are presented in comparison to the real distribution within the phantom in Table 5
Table 6 and Table 7. As previously explained, the purpose of this work is to reconstruct the changes within the conductive phase rather than finding the interface between the liquid and gas phases. The distinct boundaries in the localised reconstructed images, i.e., in ROI columns, are used to mark the ROI area, above which the conductivity changes were set to zero. 

In Table 5, the ROI images do not show an obvious improvement in image quality as the water level is relatively high and not much information is missing due to the exclusion of erroneous measurements. However, a notably better shape preservation of objects can be observed from the ROI images in Table 6 and Table 7 as opposed to the global images. Additionally, as the objects move from the edge (P1) to the centre (P3) of the pipe in each table, both reconstruction methods tend to generate severely distorted images. Yet, with the global images in P3 rows of Table 7 inaccurately spread beyond the conductive area, which would massively mislead the information processing, the ROI method brings the robustness by reliably localising the objects. 

Multiple inclusion test

Another set of tests were carried out with more than one sample in the tank and the results are presented in Table 8. As the small object tends to generate low amplitude responses, we placed the small object closer to the boundary of the phantom to simplify the problem. As stated before, when the water level is sufficiently high, which is Level 1 in our case, applying the localised method does not make an impressive difference to the image qualities. This can also be confirmed in the multiple sample test in the Level 1 row of Table 8. It is also notable that in the Level 1 simulation, both global and ROI methods struggled to separate these two inclusions. This is due to the fact that one inclusion is placed next to the boundary whereas the other is closer to the centre. Nevertheless, the better distinguishability of two objects brought by ROI method can be noted in the Level 3 test. 

#### 5.1.3. Image Analysis of Single Inclusion Experiments

To further quantitatively analyse the effect of applying limited region method in the single inclusion tests, four evaluation parameters were introduced. Due to the complexity of the images and image reconstructions involved in ERT problems, we introduced two sets of evaluation parameters to make a comprehensive inspection. Firstly, we adopted three figures of merits defined in [19], which focus on the quality of targets, namely position error (PE), shape deformation (SD), and amplitude response (AR). The significance of these parameters in the wastewater flow applications were discussed in [19]. Secondly, three additional parameters were introduced to make judgements on the overall performance of reconstructions, including the correlation coefficient (CC), relative error (RE), and computational time (CT). 

Each reconstructed image is comprised of 50 × 50 pixels and for a better accuracy of the evaluation parameters, images are resized to 200 × 200 pixels and can be represented by a column vector $\hat{x}$. In the reconstructed images ${\hat{x}}_{q}$, a threshold of one-fourth of the maximum amplitude is applied, which detects most of the visually significant effects: (4)$$[{\hat{x}}_{q}{]}_{i}=\begin{array}{c} \left\{\begin{array}{cc} 1, & \mathrm{if}\;[{\hat{x}}_{q}{]}_{i}\;\geq \;\frac{1}{4}\max(\hat{x})\\ \;\;0, & \mathrm{otherwise} \end{array}\right. \end{array}.$$
Position error

In Figure 6, the position errors of using two methods are plotted against various locations in all three level cases. The PE plot of the medium object at Location 1 in Figure 6a does not suggest a significant improvement by using the localised method. As the liquid level goes down and the objects are placed further away from the boundary, PEs see a notable increase with all reconstruction mechanisms. However, applying the localised algorithm manages to lower the PEs for both small and medium objects when compared with the images reconstructed using traditional global method.
Shape deformation

The shape deformation is compared between the global and limited region methods in Figure 7. An increase in SD for both methods is seen as the objects get further away from the boundary, which again confirms the reconstruction difficulty due to the ill-posed nature of ERT problems. The lower value of SD produced by the ROI reconstruction, especially in the cases of lower water level cases (in Figure 7b,c), however, can confirm a better preservation of using such method.
Amplitude response

In Figure 8, the sizes of the objects are known and the AR of the reconstructed images using the global and localised methods can be compared against the real distribution. In each plot, the theoretical amplitude response is plotted in dashed lines as a reference. The reconstructions using the localised method perform better than those using the global method as ARs of ROI images are closer to the corresponding theoretical values.
Pearson’s correlation coefficient

Correlation coefficient compares the similarity of the reconstructed images to the real distribution. For two grayscale images ${\hat{x}}_{q}$, ${\hat{x}}_{0}$, the correlation is defined by: (5)$${CC}_{{\hat{x}}_{q},{\hat{x}}_{0}}\;=\;\frac{cov({\hat{x}}_{q},{\hat{x}}_{0})}{\sigma_{{\hat{x}}_{q}}\sigma_{{\hat{x}}_{0}}},$$
where cov is covariance, and $\sigma_{X}$, $\sigma_{Y}$ are standards deviations of the pixel values. The closer CC is to 1, the more similar the reconstructed image ${\hat{x}}_{q}$ is to the real images ${\hat{x}}_{0}$.

The plots of CC of the experimental tests under three water levels are presented in Figure 9. As discussed before, a low water level, the close position to the centre of inclusions and a small inclusion size could lead to the difficulty of image reconstruction. This can also be observed as a decrease in correlation coefficient between the reconstructed and the true images. Moreover, the CCs of ROI images are generally smaller than those of global images, which advises an improvement made by the limited region reconstruction method. 

Relative error

Relative error measures the difference between the reconstructed images and the real images with respect to the real images. It can be defined as
(6)$$\mathrm{RE}=\frac{∥{\hat{x}}_{0}\;-\;{\hat{x}}_{q}∥}{∥{\hat{x}}_{0}∥}$$

As suggested from the definition of RE, a smaller value indicates a better reconstruction quality. Figure 10 compares the RE plots of the reconstructed images using global and limited region methods. The advantages of applying the proposed method become noticeable when fewer valid measurements are available. As mentioned previously, RE, as well as CC, is introduced to assess the overall reconstruction performances rather than only the target qualities. However, as the demonstrations were set up with one inclusion for simplification, the overall performance of the reconstructions agrees with the target qualities. This can be confirmed by lower REs and Higher CCs offered by ROI mechanism in Figure 9 and Figure 10. That is to say, the CCs and REs manifest a better recovery of the real images offered by using ROI method as opposed to the global method, as suggested in PE, SD and AR analysis. 

Computational time

The computation time is the time required for executing the image reconstruction. As we introduced FEM to simplify the continuous problem into a discretized problem, the image reconstruction is concretely a matrix calculation problem in practice. In the limited region method, the sensitivity matrices involve fewer elements when the size of individual element remains the same; and hence it will spend less time in the mathematical calculation. The computation time (in seconds) taken for recovering small and medium objects in the three water level tests is listed below in Table 9. 

As discussed before, in both the medium and small object tests, the computation time of the limited region method is always smaller than that of global region method. It is also worth noting that, as the water level drops, the ROI area shrinks accordingly, which results in a shorter computation time. This also offers an opportunity for using the localised method to increase the spatial resolution with a finer segmentation under the same computation time. 

## 6. Conclusions 

This research has added an important new feature to the previously developed ERT system for wastewater monitoring into serving part full horizontal pipes. An electrode sensing module has been introduced to determine the instructive electrodes and hence extract the valid measurements, and a dedicated reconstruction method that restricts the reconstruction area to the region of interest (ROI), which is the conductive section within the pipe, is employed to improve the robustness of the system. The impact of this work is to provide a solution for recovering the additional information on local concentration profiles within the flow. The proposed localised method was validated by both simulation and experiments under the different levels of flow in comparison with the conventional method. Both profiles have been compared to the real distributions graphically and quantitatively; and it suggests a better agreement between the localised profile and the reality. Therefore, the limited region ERT can be employed to compensate for the system’s weak immunity to noise resulting from the incomplete data. 

## Figures and Tables

**Figure 1 sensors-20-01899-f001:**
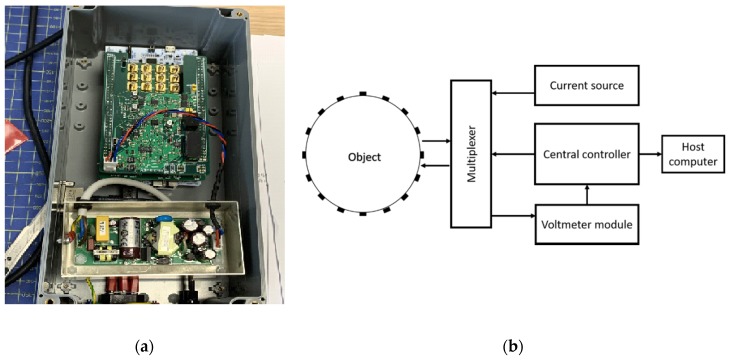
(**a**) 16-channel electrical resistance tomography (ERT) device designed in [18]; (**b**) ERT system overview.

**Figure 2 sensors-20-01899-f002:**
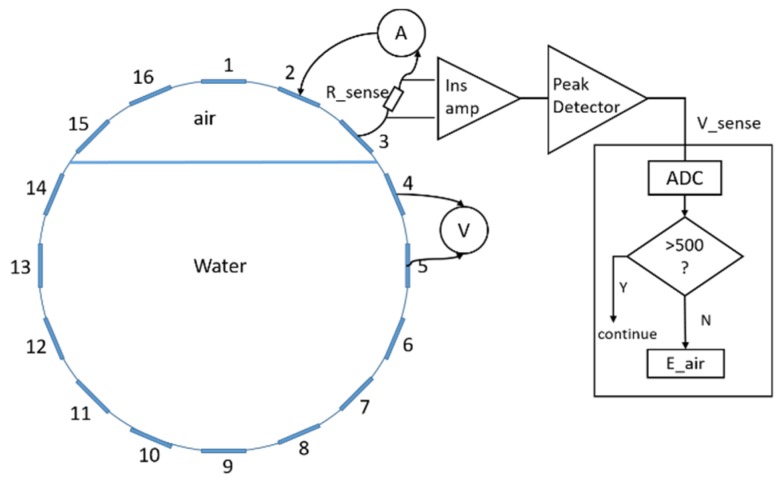
Current sensing module.

**Figure 3 sensors-20-01899-f003:**
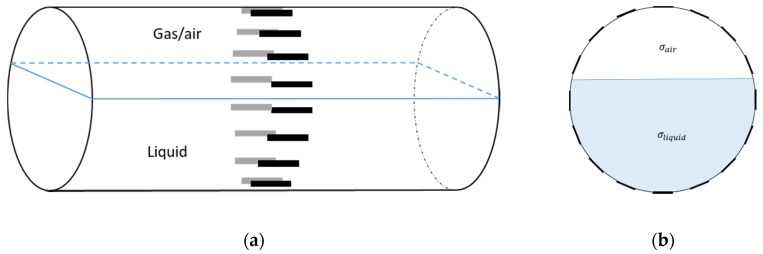
Gas-liquid ERT system model: (**a**) gas-liquid stratified flow in a horizontal pipe; (**b**) cross-sectional distribution of ERT system.

**Figure 4 sensors-20-01899-f004:**
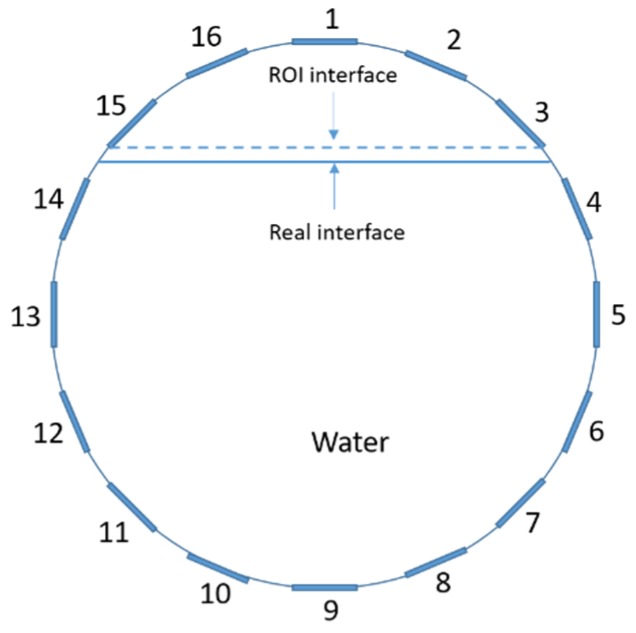
Region of interest (ROI) definition in a partially filled pipe.

**Figure 5 sensors-20-01899-f005:**
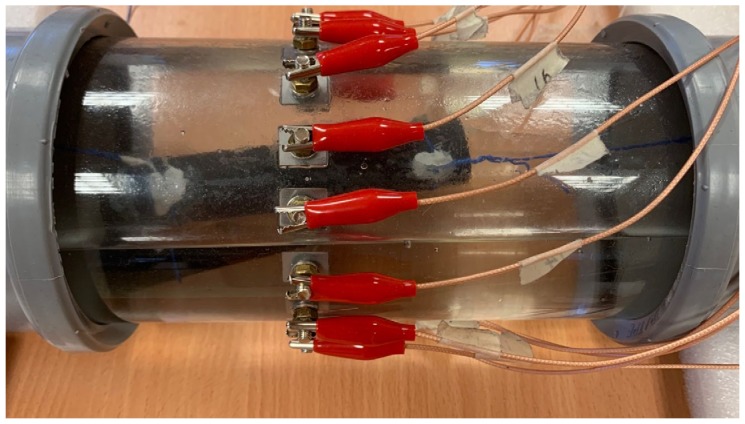
Phantom set up.

**Figure 6 sensors-20-01899-f006:**
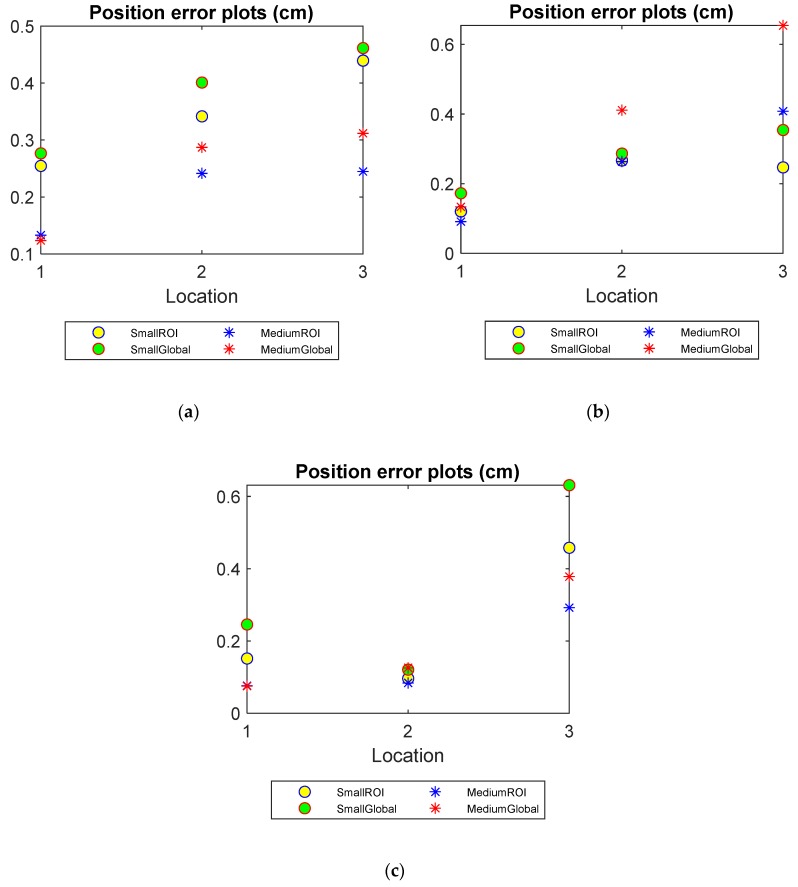
Position errors (PE) of small and medium objects inserted into the pipe filled up to three water levels: (**a**) Level 1 (**b**) Level 2 (**c**) Level 3.

**Figure 7 sensors-20-01899-f007:**
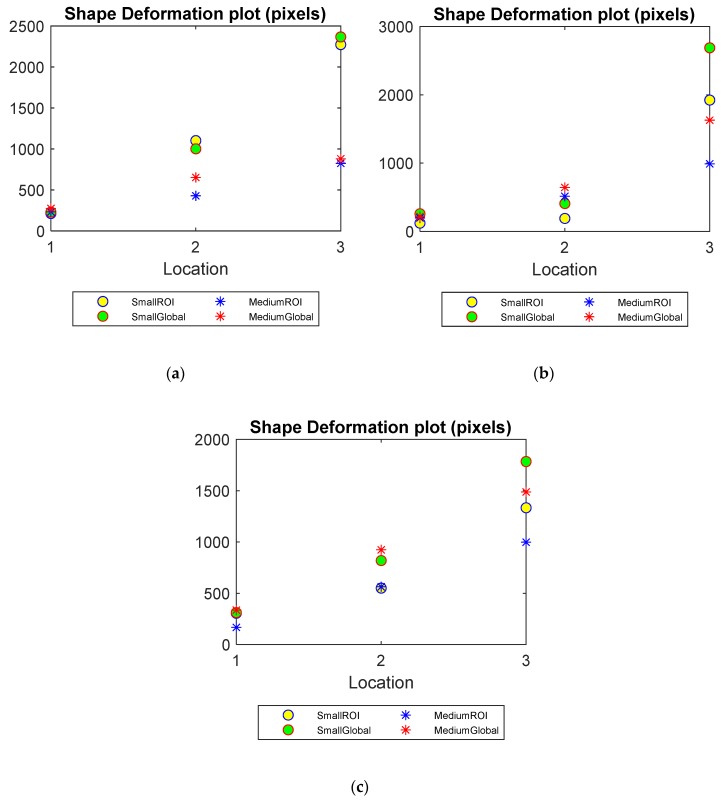
Shape deformation (SD) of small and medium objects inserted into the pipe filled up to three water levels: (**a**) Level 1 (**b**) Level 2 (**c**) Level 3.

**Figure 8 sensors-20-01899-f008:**
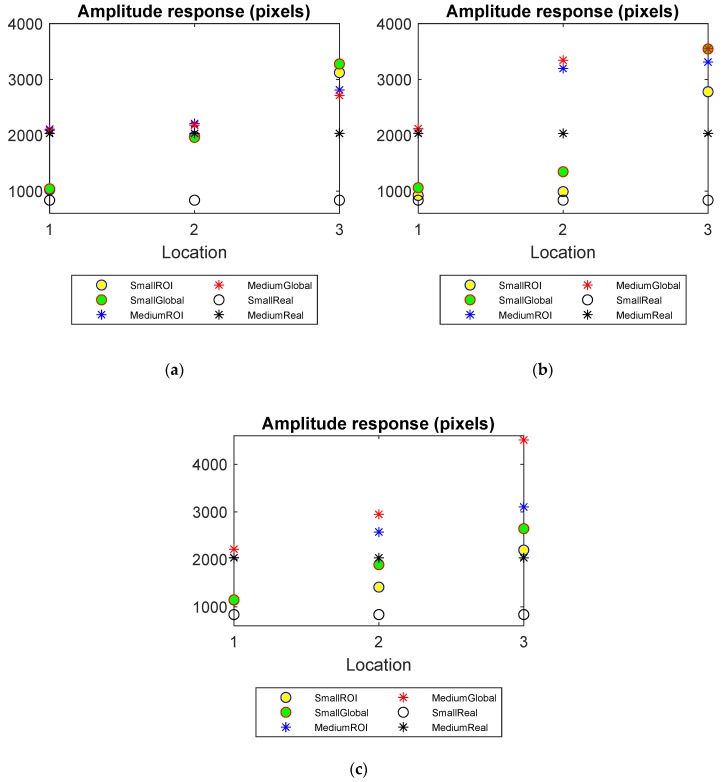
Amplitude response (AR) of small and medium objects inserted into the pipe filled up to three water levels: (**a**) Level 1 (**b**) Level 2 (**c**) Level 3.

**Figure 9 sensors-20-01899-f009:**
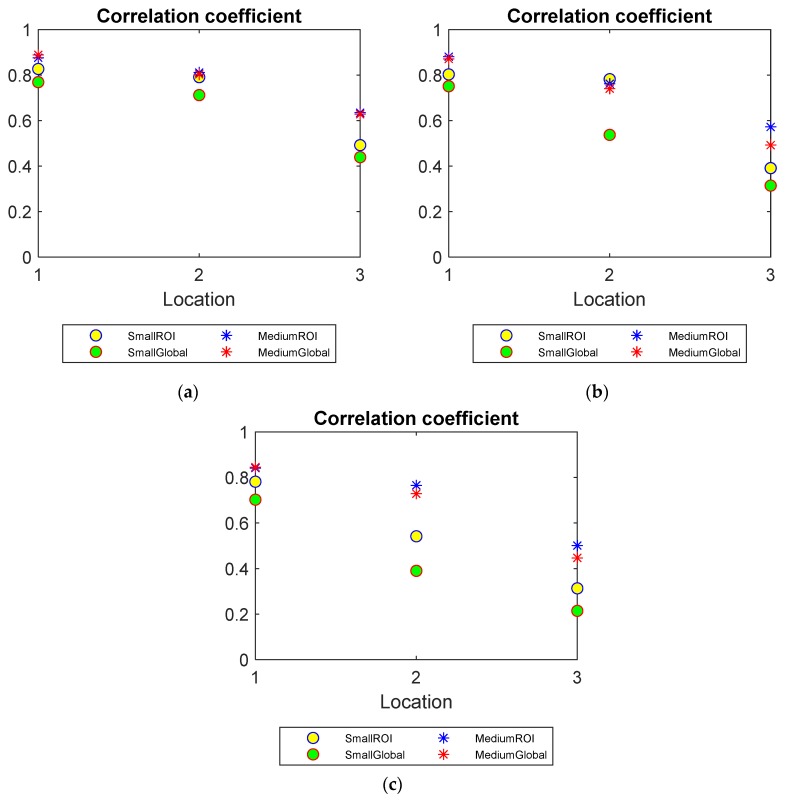
Correlation coefficient (CC) of small and medium objects inserted into the pipe filled up to three water levels: (**a**) Level 1 (**b**) Level 2 (**c**) Level 3.

**Figure 10 sensors-20-01899-f010:**
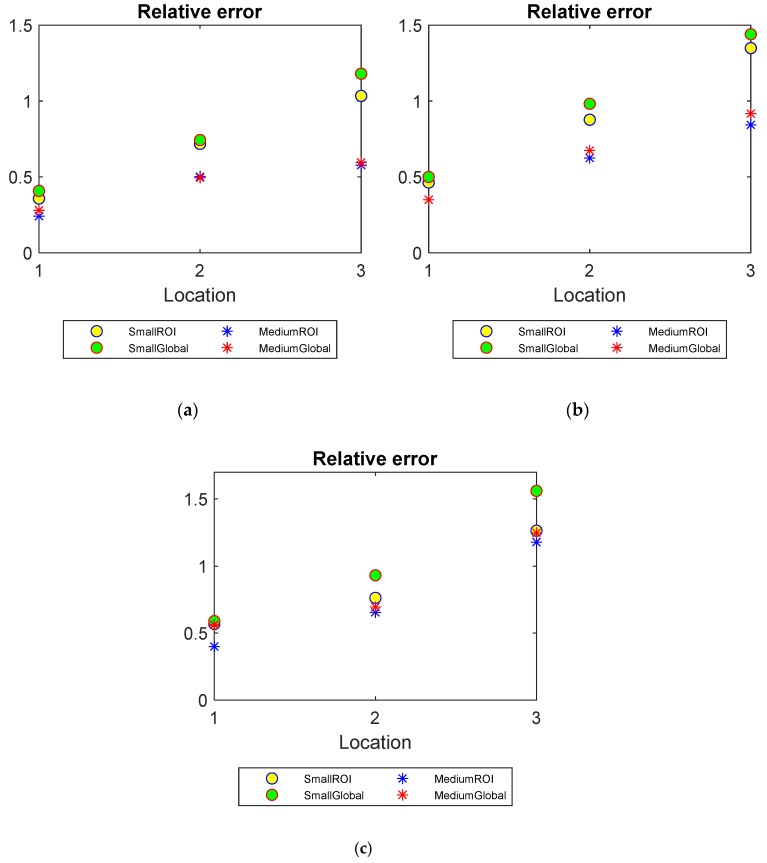
Relative error (RE) of small and medium objects inserted into the pipe filled up to three water levels: (**a**) Level 1 (**b**) Level 2 (**c**) Level 3.

**Table 1 sensors-20-01899-t001:** Single inclusion simulation results.

	Case 1	Case 2	Case 3
	Without Noise	With 12 dB Noise	Without Noise	With 12 dB Noise	Without Noise	With 12 dB Noise
Localized	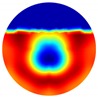	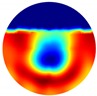	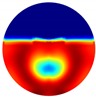	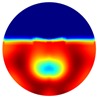	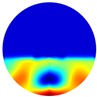	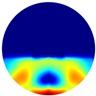
Global	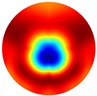	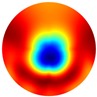	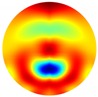	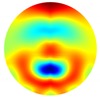	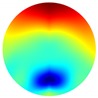	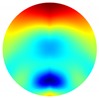
Real	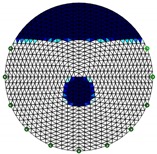	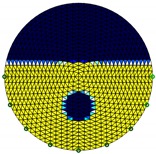	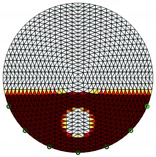

**Table 2 sensors-20-01899-t002:** Multiple inclusion simulation results.

Level 1	Level 2
Real Distribution		Image	Real Distribution		Image
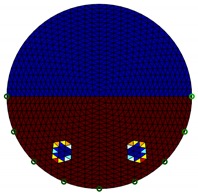	**Localised**	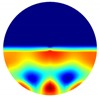	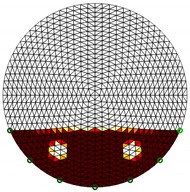	**Localised**	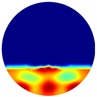
**Global**	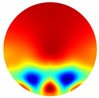	**Global**	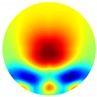

**Table 3 sensors-20-01899-t003:** Small inclusion tests comparison.

	Level 1	Level 2	Level 3	Level 4	Level 5
**Real**	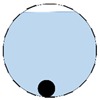	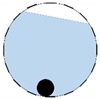	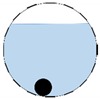	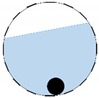	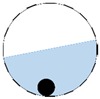
**Full Dataset**	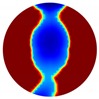	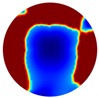	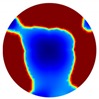	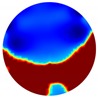	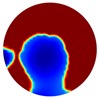
**Valid Dataset**	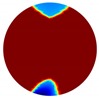	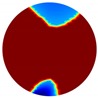	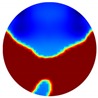	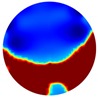	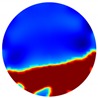

**Table 4 sensors-20-01899-t004:** Medium inclusion test comparison.

	Level 1	Level 2	Level 3	Level 4	Level 5
**Real**	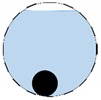	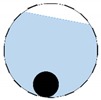	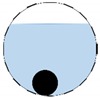	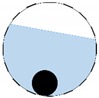	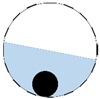
**Full Dataset**	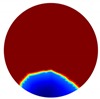	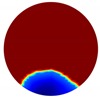	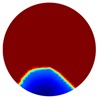	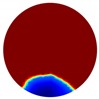	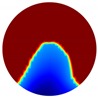
**Valid Dataset**	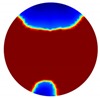	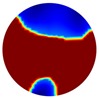	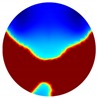	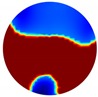	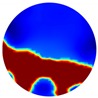

**Table 5 sensors-20-01899-t005:** Reconstructed images of small and medium objects in Level 1 tests.

	Small	Medium
	Real	Global	ROI	Real	Global	ROI
**P 1**	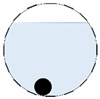	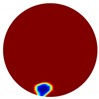	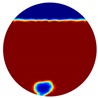	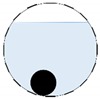	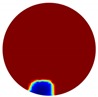	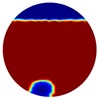
**P 2**	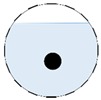	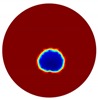	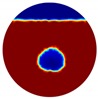	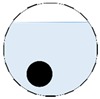	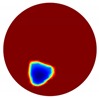	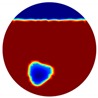
**P 3**	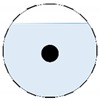	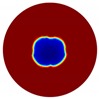	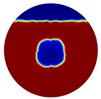	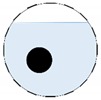	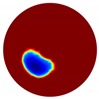	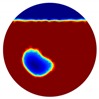

**Table 6 sensors-20-01899-t006:** Reconstructed images of small and medium objects in Level 2 tests.

	Small	Medium
	Real	Global	ROI	Real	Global	ROI
**P 1**	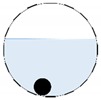	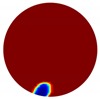	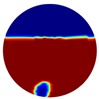	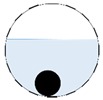	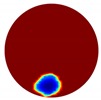	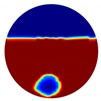
**P 2**	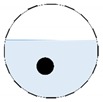	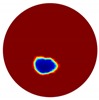	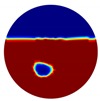	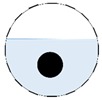	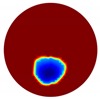	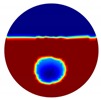
**P 3**	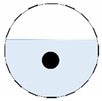	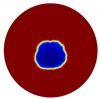	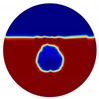	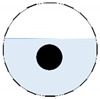	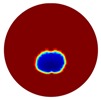	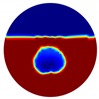

**Table 7 sensors-20-01899-t007:** Reconstructed images of small and medium objects in Level 3 tests.

	Small	Medium
	Real	Global	ROI		Global	ROI
**P 1**	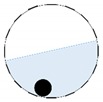	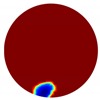	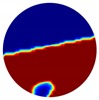	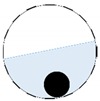	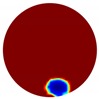	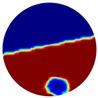
**P 2**	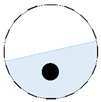	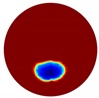	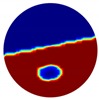	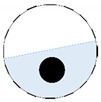	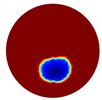	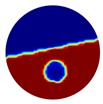
**P 3**	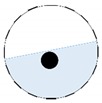	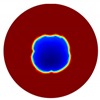	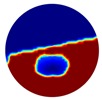	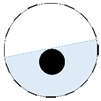	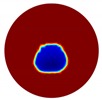	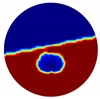

**Table 8 sensors-20-01899-t008:** Reconstructed images of multiple samples using global and ROI methods.

	Real	Global	ROI
**Level 1**	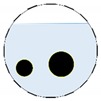	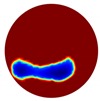	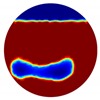
**Level 2**	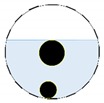	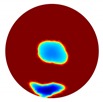	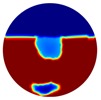
**Level 3**	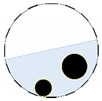	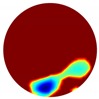	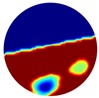

**Table 9 sensors-20-01899-t009:** Computation time of reconstructing small and medium sample using global and localised methods.

	Medium						Small					
	Global			ROI			Global			ROI		
	P1	P2	P3	P1	P2	P3	P1	P2	P3	P1	P2	P3
**Level 1**	0.45	0.47	0.50	0.30	0.38	0.37	0.49	0.48	0.49	0.38	0.36	0.34
**Level 2**	0.42	0.44	0.47	0.23	0.26	0.30	0.42	0.43	0.42	0.23	0.23	0.22
**Level 3**	0.43	0.43	0.42	0.18	0.19	0.22	0.44	0.45	0.41	0.19	0.21	0.17

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
