# Peer review of "Limited Angle Electrical Resistance Tomography in Wastewater Monitoring"

_sensors, 2020, doi:10.3390/s20071899_

Round 1
Reviewer 1 Report
Authors should remove equation (1). It is unnecessary (it does not have to be compact in every article by the same authors!). In sentences 142-144, information is necessary for the reader about where further mathematical divagations come from. I believe that the other amendments introduced meet the minimum requirements for the manuscript to appear as an article. Please improve the bibliography formatting.
Author Response
Authors should remove equation (1). It is unnecessary (it does not have to be compact in every article by the same authors!). In sentences 142-144, information is necessary for the reader about where further mathematical divagations come from. I believe that the other amendments introduced meet the minimum requirements for the manuscript to appear as an article. Please improve the bibliography formatting.
Answer: Thanks, both done.
Reviewer 2 Report
In this paper, the authors have presented a modified ERT system that has the capacity to reconstruct inclusions within the conductive region using limited data. This research is of great significance in wastewater monitoring. The English writing is well and good performance has been demonstrated. So I would recommend this paper for publication.
Author Response
In this paper, the authors have presented a modified ERT system that has the capacity to reconstruct inclusions within the conductive region using limited data. This research is of great significance in wastewater monitoring. The English writing is well and good performance has been demonstrated. So I would recommend this paper for publication.
Answer: Thanks.
Reviewer 3 Report
This paper reports an efficient design of ERT system for wastewater metering. In the partially filled lateral pipes, the electrodes lost contact with the conductive medium and, thus, incomplete data is measured. The presented design is capable of solve the inverse problem using such limited data. In my opinion, the presented material contains sufficient novelties and the manscript is in general very well written. It is also noted that Prof. Soleimani is a world-renowned expert in this field. So I think this paper is likely to attract many ERT researchers and make great impact. I only have a minor suggestion for the authors to consider.
Sequential ERT frames, or data collected from various tomography sensors are generally spatiotemporally correlated, which connotes redundancy among the observations. It is thus reasonable to consider exploiting such redundancy to realize efficient and robust learning of ERT frame sequence. Bayesian approach represents a natural and effective mechanism to incorporate the structure knowledge in solving the ERT inverse problem [Ref1]. Bayesian approach also shows its superiority in compressive sensing using incomplete data in other fields. Hence, I believe that combining the Bayesian approach with the design showcased in this manuscript can be promising in further improving the imaging performance.
[Ref1] Time Sequence Learning for Electrical Impedance Tomography Using Bayesian Spatiotemporal Priors, IEEE Transactions on Instrumentation and Measurement, vol. 69, in press, (DOI: 10.1109/TIM.2020.2972172), 2020.
Author Response
This paper reports an efficient design of ERT system for wastewater metering. In the partially filled lateral pipes, the electrodes lost contact with the conductive medium and, thus, incomplete data is measured. The presented design is capable of solve the inverse problem using such limited data. In my opinion, the presented material contains sufficient novelties and the manscript is in general very well written. It is also noted that Prof. Soleimani is a world-renowned expert in this field. So I think this paper is likely to attract many ERT researchers and make great impact. I only have a minor suggestion for the authors to consider.
Sequential ERT frames, or data collected from various tomography sensors are generally spatiotemporally correlated, which connotes redundancy among the observations. It is thus reasonable to consider exploiting such redundancy to realize efficient and robust learning of ERT frame sequence. Bayesian approach represents a natural and effective mechanism to incorporate the structure knowledge in solving the ERT inverse problem [Ref1]. Bayesian approach also shows its superiority in compressive sensing using incomplete data in other fields. Hence, I believe that combining the Bayesian approach with the design showcased in this manuscript can be promising in further improving the imaging performance.
[Ref1] Time Sequence Learning for Electrical Impedance Tomography Using Bayesian Spatiotemporal Priors, IEEE Transactions on Instrumentation and Measurement, vol. 69, in press, (DOI: 10.1109/TIM.2020.2972172), 2020.
Answer: Very relevant and great paper we have added to the reference.